MAPK/ERK signaling pathway in rheumatoid arthritis: mechanisms and therapeutic potential

Xie Jun 1 2
Sun Sijuan 1
Li Qingzhou 1
Chen Yuhui 3
Huang Lijun 3
Wang Dong 3 dwang@cdutcm.edu.cn
Wang Yumei 3 yumeiwang@cdutcm.edu.cn
1 State Key Laboratory of Southwestern Chinese Medicine Resource, School of Pharmacy, Chengdu University of Traditional Chinese Medicine , Chengdu , China
2 Santai People’s Hospital , Mianyang , China
3 School of Basic Medical Sciences, Chengdu University of Traditional Chinese Medicine , Chengdu , China
Upadhyay Rohit
Electronic publication date: 2025 Jul 14
Publication date: 2025
Volume: 13
Electronic Location ID: e19708
Received 2025 Mar 4; Accepted 2025 Jun 16
Copyright: © 2025 Xie et al.
Copyright year: 2025
Copyright holder: Xie et al.
License: This is an open access article distributed under the terms of the Creative Commons Attribution License, which permits unrestricted use, distribution, reproduction and adaptation in any medium and for any purpose provided that it is properly attributed. For attribution, the original author(s), title, publication source (PeerJ) and either DOI or URL of the article must be cited.
License URL: https://creativecommons.org/licenses/by/4.0/

Keywords: MAPK/ERK, Rheumatoid arthritis, Inflammatory response, Cell proliferation, Cell migration, Angiogenesis

Funding: National Natural Science Foundation of China 82172723 and 82304887 Ministry of Science and Technology of China through the National Key R&D Plan 2023YFF0720300 Sichuan Province Science and Technology Support Program 2024YFFK0156 Sichuan Provincial Administration of Traditional Chinese Medicine 2024MS170 Innovation Team and Talents Cultivation Program of National Administration of Traditional Chinese Medicine ZYYCXTD-D-202209 This work was supported by the National Natural Science Foundation of China (No. 82172723, No. 82304887), the Ministry of Science and Technology of China through the National Key R&D Plan (No. 2023YFF0720300), Sichuan Province Science and Technology Support Program (No. 2024YFFK0156), Sichuan Provincial Administration of Traditional Chinese Medicine (No. 2024MS170), the Innovation Team and Talents Cultivation Program of National Administration of Traditional Chinese Medicine (No. ZYYCXTD-D-202209). The funders had no role in study design, data collection and analysis, decision to publish, or preparation of the manuscript.

==============================
Rheumatoid arthritis (RA) is a multifaced autoimmune disorder characterized by chronic joint inflammation, leading to progressive disability and significantly impacting patients’ quality of life. Despite advances in treatment, finding a cure or preventing disease progression remains a major clinical challenge, underscoring the urgent need for novel therapeutic strategies. Among various pathways involved in the pathophysiology of RA, the mitogen-activated protein kinases/extracellular regulated protein kinases (MAPK/ERK) pathway is of particular importance. As the central cascade within the broader MAPK signaling pathways, MAPK/ERK plays a critical role in regulating numerous physiological and pathological processes, with a well-established and prominent involvement in RA. Unlike p38 MAPK and c-Jun-N-terminal kinase (JNK), whose role in RA have been well-documented, the specific contributions of the MAPK/ERK pathway to RA remains comprehensively unreviewed. Furthermore, the MAPK/ERK pathway does not act in isolation but interacts synergistically with other major pathways, including NF-κB, Janus kinase-signal transducer and activator of transcription (JAK/STAT), sonic hedgehog (SHH), and PI3K/AKT, which further enhance its pathological effects. This review offers a comprehensive analysis of MAPK/ERK signaling pathway, focusing on its molecular components and its contribution to RA pathophysiology. Furthermore, we explore the cross-talk between MAPK/ERK and other pathways in the context of RA, and evaluates the therapeutic potential of targeting this pathway with small molecule inhibitors, natural compounds and biomolecules. By elucidating the mechanistic role of MAPK/ERK in RA, this article aims to highlight the pathway’s therapeutic relevance and provide a foundation for the developing more effective, targeted therapies for RA.

Introduction

Rheumatoid arthritis (RA) is a debilitating autoimmune disorder characterized by chronic, recurrent inflammation affecting multiple joints (Smolen, Aletaha & McInnes, 2016). The primary clinical manifestations of RA include joint swelling and pain. In advanced stages, joint destruction and deformities may occur, significantly impairing patients’ mobility. Epidemiological data indicate that approximately 24 million people worldwide are affected by RA, with a higher prevalence among individuals aged 40 to 60 years, and a marked gender disparity, as women are approximately three times more likely to develop RA than men (Ngo, Steyn & McCombe, 2014; Rufino et al., 2024). The pathogenesis of RA is driven by a combination of inflammation processes, excessive cell proliferation, invasion and migration, and angiogenesis, all of which contribute to the progressive deterioration of the joints (Ma et al., 2019; Meng et al., 2024). Despite advancements in the treatment of RA, current therapies predominantly aim to manage symptoms and slow disease progression, but they are often limited by factors such as suboptimal efficacy, high cost, and considerable variability in individual responses (Smolen, Aletaha & McInnes, 2016). Consequently, achieving a cure or preventing disease progression remains a significant clinical challenge. Therefore, a comprehensive grasp of the molecular mechanisms driving RA, particularly the pivotal signaling pathways implicated, is essential for refining therapeutic approaches and advancing the creation of more precise and efficacious targeted therapies.

Extensive research has elucidated the involvement of various pathways in the pathogenesis of RA, including phosphoinositide 3-kinase/protein kinase B (PI3K/AKT) (Ba et al., 2021), janus kinase/signal transducer and activator of transcription (JAK/STAT) (Hu, Liu & Zhang, 2022), mitogen-activated protein kinase (MAPK) (Cai, Deng & Yao, 2024), Wnt/β-Catenin (Guo et al., 2023), and nuclear transcription factor-κB (NF-κB) (van Loo & Beyaert, 2011). These pathways collectively contribute to the inflammatory and degenerative processes central to RA. Among them, the MAPK pathway stands out as one of the most frequently dysregulated in RA, rendering it a promising target for therapeutic intervention (Behl et al., 2021). Although the MAPK family includes other members such as p38 and c-Jun-N-terminal kinase (JNK), the MAPK/ERK pathway is considered one of the most classical and central pathways within the MAPK family (Guo et al., 2020). Unlike the p38 MAPK and JNK pathways, which are more closely associated with cellular stress responses and programmed cell death (Zou et al., 2016; Cheng et al., 2022b), the ERK pathway plays a dominant role in cell growth, differentiation, and survival, underscoring its importance in both normal cellular function and disease processes (Kyriakis & Avruch, 2001; Morrison, 2012; Yang et al., 2022). Furthermore, while the roles of the p38 and JNK pathways in RA have been well documented (Schett, Zwerina & Firestein, 2008; Lai, Wu & Lai, 2020), the ERK pathway’s contribution to RA has yet to be comprehensively reviewed. Therefore, this review focuses specifically on the MAPK/ERK cascade. Moreover, MAPK/ERK signaling does not function in isolation; it interacts synergistically with other key pathways, including JAK/STAT, NF-κB, sonic hedgehog (SHH), and PI3K/AKT, further amplifying its role in the disease process (Chen et al., 2004; Liu et al., 2018a; Aripova et al., 2023).

This review explores the role of the MAPK/ERK pathway in the pathogenesis of RA. It begins by detailing the core components of the MAPK/ERK pathway, followed by its biological functions and the pathological mechanisms that activate this pathway in RA. This review further investigates the involvement of MAPK/ERK in regulating critical processes, such as inflammation, cell proliferation, cell migration and invasion, and angiogenesis in the context of RA. Additionally, it explores the complex interaction and crosstalk between MAPK/ERK and other signaling pathways that contribute to the disease progression. Finally, the review assesses the therapeutic potential of targeting the MAPK/ERK pathway with various drugs, natural compounds, and biomolecules, providing insights into their mechanisms of action in RA management. This article aims to present a comprehensive overview of the MAPK/ERK pathway’s mechanistic involvement in RA and to highlight the potential of pathway-targeted strategies for therapeutic intervention, thereby appealing to scientists or clinicians working in rheumatology and immunology and guiding their future research directions in the identification of novel targets for RA treatment.

Survey methodology

We searched for relevant literature in PubMed using the keywords (MAPK/ERK) OR (Ras-Raf-MEK-ERK) OR (ERK) AND (rheumatoid arthritis). Emphasis was placed on articles published since 2015, but earlier articles were also included. The final selection of references included studies on the role of MAPK/ERK in RA, therapeutic strategies targeting MAPK/ERK in RA and crosstalk between MAPK/ERK and other pathways. Only research and review articles were included, while documents that were not classified as articles or reviews were excluded.

Molecular composition, activation mechanisms, and physiological functions of the MAPK/ERK pathway

Structural and functional components of the MAPK/ERK pathway

In the MAPK/ERK pathway, Ras is the initiating activator protein, while Raf and MAPK/ERK kinase (MEK) function as MAPKKK and MAPKK, respectively, and ERK is designated as MAPK, collectively establishing the Ras-Raf-MEK-ERK cascade (Yang & Liu, 2017). Ras is a small G-protein that initiates the MAPK/ERK cascade and consists of 188–189 amino acids, featuring 6 β-folded and 5 α-helical structures, along with a CAAX sequence at the C-terminus (Zhou, Der & Cox, 2016). It exists in two different conformations: a form bound to GDP that is inactive, and an active form that binds to GTP (Simanshu, Nissley & McCormick, 2017).

The Raf protein kinase family includes three isoforms: A-Raf, B-Raf, and Raf-1. consisting of approximately 600 to 670 amino acids (Hashem et al., 2024). Raf kinase comprises three conserved structural domains: CR1, CR2, and CR3. The CR1 is responsible for binding to Ras, while the CR2 contains several phosphorylation sites that are crucial for the regulation of Raf. Additionally, the CR3 serves as the kinase domain of Raf (Sithanandam et al., 1990).

MEK proteins exist in two isoforms: MEK1 and MEK2. MEK1 consists of 389 amino acids, while MEK2 comprises 400 amino acids. Both MEK proteins contain a characteristic dual-phosphorylated kinase domain and a specific ERK-binding site, which facilitates their efficient phosphorylation of ERK (Ohren et al., 2004). The ERK family is primarily represented by two core members: ERK1, consisting of approximately 379 amino acids, and ERK2, which is made up of about 360 amino acids (Boulton et al., 1991; Burkhard et al., 2009).

Mechanism of activation in MAPK/ERK pathway

The MAPK/ERK pathway can be stimulated by various extracellular stimuli and intracellular signaling molecules, including chemical toxins, ultraviolet radiation, infections by pathogens (e.g., bacteria and viruses), growth factors, and cytokines (Bruder & Kovesdi, 1997; Kim et al., 2006; Chong et al., 2023; Kuonqui et al., 2023). The Guanine Nucleotide Exchange Factor (GEF) facilitates the transition of Ras from a state bound to GDP to one bound to GTP upon cellular stimulation, leading to the activation of Ras (Mor & Philips, 2006). Ras operates as an upstream activating protein by utilizing two domains: the cysteine-rich structural domain and the Ras-binding structural domain. These regions enable Ras to bind to Raf and facilitate its translocation from the cytoplasm to the cell membrane. Raf is subsequently activated (Jeon, Tkacik & Eck, 2024). Upon activation, the C-terminal catalytic domain of Raf engages with MEK, catalyzing the phosphorylation of subregion VIII at serine residues, thus activating MEK (Santarpia, Lippman & El-Naggar, 2012). The activated form of MEK subsequently engages with ERK via its N-terminal region and catalyzes the phosphorylation of tyrosine and threonine residues in ERK, thereby leading to its activation. After being phosphorylated, p-ERK moves into the nucleus of the cell, where it influences gene transcription and various cellular functions (Wan et al., 2004) (Fig. 1).

Figure 1 The specific mechanism of MAPK/ERK pathway.

When cells are stimulated, Ras transitions from a GDP-bound state to a GTP-bound state, resulting in Ras activation. Subsequently, Ras binds to Raf, promoting Raf activation. Upon activation, Raf binds to MEK, leading to the activation of MEK. The activated form of MEK then interacts with ERK via its N-terminal region, catalyzing the phosphorylation of ERK and resulting in its activation. Following phosphorylation, p-ERK translocates to the nucleus, where it influences gene transcription and various cellular functions.

Physiological functions of the MAPK/ERK pathway

The MAPK/ERK pathway is a complex and vital cellular communication system, comprising multiple key components and steps. It is essential for the body’s immune response, inflammation, and anti-pathogen defense (Lacchini et al., 2006; Liang et al., 2024). It has been reported that, through T cell receptor-dependent activation, the MAPK/ERK pathway regulates T cell activation (Sofi et al., 2022). Additionally, it can phosphorylate the transcription factor ELK1, thereby controlling B cell activation and proliferation (Hipp et al., 2017). Furthermore, the MEK-ERK signaling activates the Cyclin D1/CDK4 complex to promote the G1/S phase transition, thereby mediating cell activation and proliferation (Li et al., 2022a). It can also phosphorylate paxillin to facilitate directed cell migration (Gao et al., 2009). In wound healing, ERK regulates the secretion of MMP-2, thereby promoting ECM remodeling (Wu et al., 2016). It also activates GLUT4 membrane translocation, enhancing insulin sensitivity and regulating glucose transport (Chen et al., 2002). Additionally, ERK specifically phosphorylates PPARγ to inhibit excessive adipocyte differentiation (Wu et al., 2022). Under oxidative stress, ERK activates the expression of antioxidant enzymes through Nrf2 nuclear translocation, thereby achieving stress adaptation and repair (Yusuf et al., 2019). In conclusion, the MAPK/ERK pathway is a highly intricate and finely regulated cascade that plays a vital role in integrating various extracellular signals and coordinating several physiological responses within the cell.

The roles of the MAPK/ERK pathway in RA

The abnormal activation of the MAPK/ERK pathway promotes RA pathogenesis through multiple dimensions. The Ras-Raf-MEK-ERK cascade serves as the driver of fibroblast-like synoviocytes (FLS) activation. In the inflammatory microenvironment, TNF-α and IL-1β activate receptor tyrosine kinases, triggering Ras-GTP binding, which subsequently recruits and phosphorylates Raf kinase. This activation leads to dual phosphorylation of MEK1/2 and ERK1/2 resulting in the transformation of FLS into an invasive phenotype and increased secretion of matrix metalloproteinases (Li et al., 2023). Additionally, ERK phosphorylation can directly enhance the activity of transcription factors NF-κB and AP-1, promoting the sustained expression of pro-inflammatory cytokines such as IL-6 and IL-8 (Yang et al., 2010). ERK-mediated upregulation of COX-2 leads to increased PGE2 secretion, which further activates the MAPK pathway through EP2/EP4 receptors (Harizi, Limem & Gualde, 2011). RANKL enhances NFATc1 transcriptional activity via the Ras-ERK axis, cooperating with MITF to promote osteoclast precursor differentiation (Li et al., 2022b). Moreover, ERK influences bone erosion by regulating the expression of TRAP and CTSK genes (Ban et al., 2023). In summary, the MAPK/ERK pathway plays a pivotal role in the pathological processes of RA. We summarized the role of MAPK/ERK in RA from the following four key aspects.

MAPK/ERK and inflammatory response

The MAPK/ERK pathway serves as a crucial regulatory mechanism in inflammation. Its overactivation is linked to the inflammation of synovial tissue and the degradation of articular cartilage. In RA joint tissues, MAPK/ERK demonstrates a degree of basal activity and spontaneous activation even in the absence of a clear external stimulus (Inaba et al., 2008). Upon stimulation by external signals, Ras, Raf, MEK, and ERK are sequentially activated (Morel et al., 2002). The highly activated MAPK/ERK stimulates the synthesis of various inflammatory mediators. In turn, these inflammatory factors further activates the MAPK/ERK pathway, creating a detrimental cycle that accelerates the development of RA (Zhang et al., 2023). Studies have demonstrated that TNFα effectively activates the MAPK/ERK pathway, which subsequently activates Schnurri-3. The activation of Schnurri-3 influences the Wnt/β-catenin pathway and up-regulates RANKL, leading to joint inflammation and bone loss (Stavre et al., 2023). Moreover, dysregulation of the MAPK/ERK pathway interferes with the balance of Th17 and Treg cells in the body, thus exacerbating the inflammatory response at the joints (Moon et al., 2014). Additionally, the MAPK/ERK influences macrophage polarization. For instance, Yan et al. (2024) demonstrated that activated ERK pathway upregulates the transcriptional activity of hypoxia-inducible factor 1α (HIF-1α), which subsequently regulates GLUT1 gene expression. Inhibition of this pathway reduces glycolysis and promotes the switching of macrophages to the M2 phenotype, thereby suppressing arthritis (Yan et al., 2024).

In RA, inhibiting the MAPK/ERK pathway has been shown to effectively mitigate the inflammatory response (Li et al., 2017; Cheng et al., 2024). In the collagen-induced arthritis (CIA) rat model, it was observed that geniposide treatment effectively suppressed the levels of p-Raf, p-MEK, and p-ERK1/2 in lymphocytes. Additionally, it reduced the production of IL-17 and IL-6, which contributed to a decrease in synovial tissue and ankle joint damage (Wang et al., 2017b). Furthermore, NR1D1 has a biological function in RA. Its activation prevents cartilage destruction by inhibiting p-JNK and p-ERK, thereby suppressing pro-inflammatory cytokines (Liu et al., 2020). Collectively, these findings suggest that MAPK/ERK activation is strongly related to the inflammation of the synovium, and that inhibiting MAPK/ERK activity may represent a promising strategy for treating RA (Fig. 2).

Figure 2 Functions of activated MAPK/ERK pathway in the progression of RA.

The activation of MAPK/ERK signaling is associated with the progression of RA, including mediating the inflammatory response, cell proliferation, cell migration and invasion, and angiogenesis.

The role of MAPK/ERK in cell proliferation

Synovial hyperplasia serves as a prominent characteristic of RA and a key contributor to joint damage. During the initial phases of RA, the immune system is abnormally activated, resulting in a massive release of cytokines from inflammatory cells in the synovium, leading to cell proliferation and activation. These proliferating cells not only increase in number, but also exhibit abnormal function, leading to excessive synthesis of extracellular matrix components. The extracellular matrix accumulates and affects the articular cartilage, which can eventually lead to bone destruction and dysfunction of the joint. In addition, hyperplastic synovium can lead to narrowing of the joint cavity, affecting the secretion of synovial fluid and the normal function of the joint (Qi-Shan et al., 2022; Yitong et al., 2023). The MAPK/ERK pathway transduces signals from extracellular stimuli and regulates cell growth (Xin et al., 2021; Tingwen et al., 2023).

The MAPK/ERK pathway influences cell proliferation by regulating cycle-related proteins. In RA, studies have shown that cell cycle proteins, such as Cdc37, can activate MAPK/ERK transcription, driving FLS from the G1 phase into the S phase. This transition promotes cell proliferation and exacerbates the progression of RA (Weiwei et al., 2023). Additionally, the activation of acid-sensing ion channel 1a (ASIC1a) raises intracellular calcium ion concentrations, which influences the phosphorylation status of cell cycle-related proteins and regulates the cell cycle (Li et al., 2013). Studies indicate that the activation of ASIC1a is influenced by ERK/MAPK pathway, further promoting the proliferation of FLS (Jingjing et al., 2021). P21 and P27 are two important CDK inhibitory proteins that block cell cycle progression by obstructing the activity of the CDK-cyclin complex. Seong-Su et al. (2009) demonstrated that the ERK inhibitor PD98059 can inhibit the growth of FLS by promoting the levels of P21 and P27. Furthermore, ERK activation enhances the expression of c-Myc, facilitating the progression of lymphocytes from the G1 phase to the S phase, thereby increasing cell proliferation and worsening the severity of arthritis (Choi et al., 2016).

Some proteins influence cell growth by regulating the MAPK/ERK pathway. RasGRP4 serves as a Ras protein activator, and when RasGRP4 is highly expressed, the Raf-MEK-ERK pathway is extremely susceptible to phosphorylation, thereby inducing FLS proliferation (Sanae et al., 2017). Dual-specificity tyrosine-regulated kinase 1A (Dyrk1A) activates ERK signaling through suppression of Spry2 (an inhibitor of the MAPK/ERK pathway), thereby promoting the growth of FLS (Guo et al., 2018). Granzyme K (GZMK) directly binds to CCL5, activating the ERK pathway and inhibiting ferroptosis and apoptosis while also promoting the proliferation of FLS (Xu et al., 2024b). However, it is noteworthy that ERK sometimes can inhibit apoptosis, for example, by phosphorylating and inhibiting the apoptosis-related protein Bad (Hafeez et al., 2016).

Regulating MAPK/ERK activity can effectively suppress synovial proliferation in RA. For instance, Kuan-Yu et al. (2019) found that microRNA (miR)-320a notably reduced the proliferation of FLS in the G0/G1 phase by blocking ERK1/2 phosphorylation, while also promoting FLS apoptosis. Hui et al. (2019) demonstrated that berberine inhibits FLS proliferation by binding to lysophosphatidic acid (LPA) and subsequently blocking LPA-mediated p-38/ERK phosphorylation. Furthermore, it has been suggested that silencing the long non-coding RNA NEAT1 enhances the expression of miR-129 and miR-204, which in turn inhibits the MAPK/ERK pathway, ultimately reducing synovitis and FLS proliferation (Jie et al., 2021). Collectively, these studies elucidate the relationship between MAPK/ERK pathway and synovial proliferation and apoptosis, indicating that regulating the MAPK/ERK activity could effectively suppress the proliferation of synovial tissue (Fig. 2).

MAPK/ERK in cell migration and invasion

In RA, cells such as FLSs and macrophages exhibit extremely strong migration and invasion capabilities, triggering a series of pathological changes that ultimately lead to joint destruction (Liu et al., 2022; Yang et al., 2024). The MAPK/ERK pathway enhances the adhesion between the extracellular matrix and cells by regulating the expression and activity of cell adhesion molecules and intercellular adhesion molecules, which subsequently induce cell migration and invasion. Cyr61 is a protein associated with cell adhesion and migration. It binds to various cell surface receptors, such as integrins, promoting interactions between cells and the extracellular matrix (Lau, 2011). Additionally, Cyr61 upregulates the activity of certain proteases secreted by cells, including matrix metalloproteinases (MMPs), which leads to the breakdown of the extracellular matrix and promotes cellular penetration and invasion of surrounding tissues (Zhai et al., 2017). The ERK1/2 inhibitor PD98059 considerably inhibited the IL-6- stimulated elevation of Cyr61, as well as the invasive and migratory abilities of FLS (Chang-Min et al., 2020). IL-1β promotes the levels of intercellular adhesion molecule-1 (ICAM-1) in FLS cells by the ERK pathway. The upregulation of ICAM-1 enhances leukocyte adhesion to FLS that have been exposed to IL-1β, facilitating the migration of leukocytes towards the synovium and exacerbating the inflammatory response (Yang et al., 2010). Another study demonstrated that TNFα-induced Mac-1 expression relies on ERK1/2, which facilitates neutrophil migration (Montecucco et al., 2008).

Furthermore, the secretion of cytokines, such as chemokines, is stimulated by MAPK/ERK, which delivers chemical signals that induce directed migration (Cheng et al., 2022a). Jun-Way et al. (2023) reported that nesfatin-1 upregulates CCL2 expression through the NF-κB, MEK/ERK, and p38. This induction enhances the polarization of M1 macrophages and facilitates the migration of monocytes. Inhibition of nesfatin-1 suppressed the MEK/ERK pathway, which subsequently reduced monocyte migration toward the inflamed area and lowered the levels of inflammatory markers. Consequently, the inflammatory response in CIA mice was alleviated, along with a reduction in joint damage (Jun-Way et al., 2023).

Additionally, certain proteins enhance cell migration by modulating the MAPK/ERK pathway. Huiyu et al. (2017) discovered that proprotein convertase subtilisin/kexin type 6 (PCSK6) activates the ERK1/2 in vitro. This activation has significant implications, as it leads to an increase in several fundamental cellular processes, including cell proliferation, and invasion proliferation in FLS (Huiyu et al., 2017). CCL25 can activate p38 and ERK by binding to CCR9, thereby promoting the infiltration of FLS and monocytes (Umar et al., 2021). These findings suggest that MAPK/ERK is involved in migration and invasion of cells in RA, and that targeting MAPK/ERK pathway could be a viable approach for managing RA (Fig. 2).

The contribution of MAPK/ERK to angiogenesis

Angiogenesis is a critical feature of RA. In RA, cytokines and coagulation factors recruit leukocytes into the synovium, inducing angiogenesis (David et al., 2021). The presence of numerous neovessels in the synovium enhances blood flow to the area, supplying essential nutrients and oxygen to the proliferating synovial cells while also facilitating the migration of additional immune cells. Vascular endothelial growth factor (VEGF) serves as a crucial regulator in the process of neovascularization and is significantly increased in individuals suffering from RA (Tsuyoshi et al., 2014). The MEK/ERK pathway increases VEGF expression, stimulates the development and movement of endothelial progenitor cell (EPC), and ultimately induces angiogenesis (Chien-Chung et al., 2021). The c-Met-MEK-ERK pathway positively regulates the function of MH7A cells. c-Met, which is highly expressed in RA, promotes the phosphorylation of ERK1/2 triggered by HGF in MH7A cells, thereby increasing the production of VEGF and MMP-3. This cascade ultimately leads to angiogenesis and bone destruction (Seiji et al., 2014).

Furthermore, modulation of the MAPK/ERK pathway effectively inhibits the migration of vascular endothelial cells, thereby suppressing angiogenesis. Rakesh, Terry & Sharmila (2009) demonstrated that the MAPK/ERK and PI3K/AKT pathways synergistically promote the expression of FOXO, which in turn promotes angiogenesis and facilitates human umbilical vein endothelial cells (HUVEC) migration and formation. Fractalkine (Fkn) promotes endothelial cell migration by activating JNK and ERK1/2, thereby inducing angiogenesis. Moreover, inhibiting the positive feedback mechanism of Fkn reduces the angiogenic capacity of endothelial cells, indicating that this feedback loop could be essential for the process of angiogenesis (Michael et al., 2010). Inhibition of MAPK/ERK pathway effectively reduce angiogenesis in RA. For instance, gleditsioside B initially blocks the ERK as well as the PI3K/AKT signaling pathway, resulting in down-regulation of MMP-2 and FAK expression. Consequently, the movement of HUVEC is suppressed, ultimately suppressing angiogenesis (Bei et al., 2013) (Fig. 2).

Crosstalk between MAPK/ERK and other signaling pathways in RA

Beyond the imbalance of the MAPK/ERK pathway, the progression of RA involves the imbalance and abnormal activation of several other pathways, for example, JAK/STAT, NF-κB, SHH, as well as PI3K/AKT pathways. The MAPK/ERK pathway engages in interactions and connections with other pathways, influencing the pathophysiological processes related to RA. This section outlines the relationship between the MAPK/ERK pathway and the aforementioned pathways in RA.

Interactions between MAPK/ERK, JAK/STAT and NF-κB pathways

In RA, the MAPK/ERK, JAK/STAT, and NF-κB pathways could be simultaneously activated, and these pathways interrelate with each other to contribute to the disease’s pathological processes. The dissociation of the IκBα/NF-κB complex are induced by the activation of STAT3, which ultimately results in the release of a significant quantity of pro-inflammatory cytokines. These inflammatory cytokines stimulate the MAPK/ERK pathway, resulting in increased cell proliferation and viability, thereby exacerbating the inflammatory response (Zheng et al., 2024). Conversely, ERK functions as a signaling molecule that initiate NF-κB (Mercer & D’Armiento, 2006). Kinases in the downstream signaling of the MAPK pathway, like ERK, phosphorylate and activate the IKK (IκB kinase) complex. Then, IκB is phosphorylated by the activated IKK complex, leading to the activiation of NF-κB to promote the transcription and translation of inflammatory factors (Chen et al., 2004). Prostaglandin E2 (PGE2) has been reported to suppress the translocation of p65 in FLS by blocking IL-1β/TNF-α-stimulated ERK activation, thereby attenuating inflammation (Gomez et al., 2005). Furthermore, MAPK/ERK enhances the production of IL-6, subsequently activating STAT3 (Mihara et al., 2012). STAT3 is prominent for the differentiation of Th17 cells, which exacerbate inflammation in RA by secreting IL-22 and IL-17. Furthermore, IL-17 stimulates the synthesis of MMP-1 in synoviocytes, resulting in joint destruction (Banerjee et al., 2017). Moreover, MAPK/ERK is closely associated with the proliferation of FLS in RA and can enhance FLS proliferation through the JAK/STAT pathway (Zhu et al., 2023). The crosstalk between the MAPK/ERK pathway, JAK/STAT pathway, and NF-κB pathway is shown in Fig. 3.

Figure 3 Crosstalk between MAPK/ERK, NF-κB and JAK/STAT signaling pathways in RA pathology.

MAPK and SHH pathway crosstalk

Many research efforts have shown that the SHH pathway influences cell growth, movement, and invasion by mediating the MAPK/ERK cascade (Lu et al., 2012; Ertao et al., 2016). In RA, the SHH pathway contributes to the tumor-like characteristics of smoothened (SMO)-dependent FLS (Zhu et al., 2017). The activation of SMO triggers downstream signaling positively by inducing Gli transcription factors to bind to DNA, thereby activating gene expression. The activation of MEK-1 synergizes with the traditional SHH pathway, leading to a notable increase in Gli-dependent transcriptional activity (Riobo, Haines & Emerson, 2006). Furthermore, the unusual activation of the SHH pathway, in conjunction with functional K-Ras mutations, triggers the Raf/MEK/ERK signaling cascade (Liu et al., 2018a). This regulates the cell cycle of FLS and promotes proliferation. The MAPK/ERK pathway is activated by SHH pathway through phosphorylation of two specific ERK1/2 residues, converting inactive ERK1/2 into an active form (Lu et al., 2012). In summary, these two signaling pathways are intertwined and collectively mediate the growth and migration of FLS. The crosstalk between the MAPK/ERK pathway and SHH pathway is shown in Fig. 4.

Figure 4 Crosstalk between MAPK/ERK and SHH signaling pathways in RA pathology.

The crosstalk between MAPK/ERK and PI3K/AKT pathways

There is a clear crosstalk between the PI3K/AKT and MAPK/ERK pathways. Both of them may be activated by pro-inflammatory cytokines. The activation further promotes the production of inflammatory factors. Thus, the two pathways mutually influence each other (Liu et al., 2014; Sun et al., 2020). Additionally, the activity of certain transcription factors, such as HIF-1α, is co-regulated by the MAPK/ERK and PI3K/AKT pathways. HIF-1α interacts with the HRE promoter to modulate VEGF expression, which in turn promotes angiogenesis and inflammatory responses, thereby exacerbating the pathological changes in the synovium (Park et al., 2015). The crosstalk between the MAPK/ERK pathway and PI3K/AKT pathway is shown in Fig. 5.

Figure 5 Crosstalk between MAPK/ERK and PI3K/AKT signaling pathways in RA pathology.

Therapeutic strategies targeting the MAPK/ERK in RA

As research into the pathological processes of RA deepens, several key signal transduction pathways have increasingly garnered the attention of researchers. Among these, the MAPK/ERK pathway is one of the most frequently dysregulated pathways observed in RA. Therapeutic strategies that target this pathway have gained prominence in RA research. Studies indicate that various drugs or agents can prevent or alleviate RA symptoms to some extent by modulating MAPK/ERK activity in relevant preclinical models (Zhou et al., 2022; Chen et al., 2023; Yin et al., 2024).

Small molecule inhibitors of MAPK/ERK

A variety of drugs have been shown to offer protection against RA. Atorvastatin (AT) is a statin with anti-tumour, lipid-lowering and immunomodulatory effects (Hu et al., 2021; Shaghaghi et al., 2022; Sun et al., 2022). AT inhibits TNF, IL-17A, IL-10, and IL-6 in PBMC of RA patients (de Oliveira et al., 2020). By inhibiting ERK activation, AT modulates the abnormal number of Treg cells and restores their immunosuppressive function in RA patients, thereby attenuating uncontrolled inflammation in the joints (Tang et al., 2011). Although AT has been shown to treat a variety of autoimmune diseases, long-term use may lead to adverse effects, including elevated muscle enzymes, abnormally elevated liver enzymes, and gastrointestinal reactions (Zhou et al., 2024). Therefore, regular blood lipid and liver function tests are required to monitor efficacy and safety during the medication period.

Etanercept, a TNFα inhibitor, is a biologic medication utilized for managing RA (Atzeni et al., 2021). In monocytes, etanercept has been shown to inhibit the phosphorylation of p38, JNK, and ERK, consequently reducing CCL2 expression. The therapeutic mechanisms of adalimumab and etanercept are similar (Yi-Ching et al., 2017). Due to their high cost, complex dosing regimens, and potential adverse effects (Matsui et al., 2017; Küçükali et al., 2024), it is crucial to optimize the formulation process, develop novel delivery technologies, develop biosimilars, and enhance safety monitoring.

Dexamethasone consistently upregulates MKP-1 and downregulates phosphorylation of JNK, p38, and ERK in FLS, thereby inhibiting FLS activation (Toh et al., 2004). However, prolonged use of dexamethasone diminishes its efficacy and is linked to significant adverse effects (Toroghi et al., 2022). Therefore, the adverse effects of dexamethasone can be effectively prevented and controlled through a combination of measures, such as optimization of dosing regimen, monitoring and adjuvant therapy.

Furthermore, a number of other drugs have been identified to improve RA by obstructing the function of the MAPK/ERK pathway, including quetiapine, niclosamide, and paroxetine. In CIA mouse model, quetiapine enhances the levels of anti-inflammatory factors while decreasing the levels of pro-inflammatory factors by interfering with ERK phosphorylation (Pan et al., 2018). Niclosamide inhibits pro-inflammatory factors and the migration and invasion of FLS by blocking p-ERK (Liang et al., 2015). Paroxetine suppresses chemokine production by inhibiting the ERK pathway, which in turn inhibits T-cell activation and migration into synovial tissue and protects joints from damage (Wang et al., 2017a) (Table 1).

Table 1 Small molecule inhibitors of MAPK/ERK.

Small molecule	In vivo/in vitro model	Mechanism	Function	Reference	
Atorvastatin	T cell	Inhibit ERK activation, modulates the abnormal number of Treg cells and restores their immunosuppressive function	Attenuate uncontrolled inflammation	Tang et al. (2011)	
Etanercept	Monocyte	Inhibit the phosphorylation of p38, ERK, and JNK, and reduce the production of CCL2	Relieve inflammation	Yi-Ching et al. (2017)	
Adalimumab	Monocyte	Inhibit the phosphorylation of p38, ERK, and JNK and reduce the production of CCL2	Relieve inflammation	Yi-Ching et al. (2017)	
Dexamethasone	FLS	Upregulate MKP-1 and downregulate phosphorylation of ERK, JNK, and p38	Inhibit FLS activation	Toh et al. (2004)	
Quetiapine	CIA mice	Interfere with ERK phosphorylation	Relieve inflammation	Pan et al. (2018)	
Niclosamide	FLS	Inhibit ERK phosphorylation and the secretion of pro-inflammatory factors	Anti-inflammatory and inhibit the migration and invasion of FLS	Liang et al. (2015)	
Paroxetine	T cell	Inhibit the ERK pathway and suppress chemokine production	Inhibit T-cell activation and migration into synovial tissue and protect joints from damage	Wang et al. (2017a)	

Natural compounds modulating MAPK/ERK activity

Although the aforementioned drugs have been shown to improve RA, they are associated with unavoidable drawbacks, including side effects, high costs, and prolonged treatment durations (Tarp et al., 2017). Consequently, there is an increasing interest in natural products as alternatives for alleviating RA and related inflammatory diseases (Dudics et al., 2018).

In RA, abnormal activation of FLS is manifested by hyperproliferation and abnormal migration. This ultimately leads to hyperplasia of the synovium and the degradation of bone (Yan et al., 2021). Therefore, targeting the activity of FLS may be an effective treatment for RA. Research has demonstrated that imperatorin can ameliorate RA by obstructing the MAPK/ERK, thereby inhibiting the excessive activation of FLS (Wei et al., 2022). Furthermore, naringenin treatment has been shown to impede the phosphorylation of ERK and Akt in FLS, which in turn reduces inflammation and MMPs production while promoting apoptosis in FLS (Yirixiati et al., 2021). Sinomenine (SIN), an alkaloid extracted from Sinomenium acutum, possesses anti-tumor, sedative, and anti-inflammatory properties (Liu et al., 2018b; Jiang et al., 2024). α7nAChR is directly implicated in the proliferation of FLS (Yi et al., 2018). The early growth response gene-1 (Egr-1) serves as a transcription factor for the gene encoding α7nAChR. Studies have demonstrated that SIN acts as a potent inhibitor of α7nAChR. In RA, SIN mitigates synovial inflammation and FLS proliferation, potentially by decreasing the protein level of Egr-1 through the inhibition of ERK1/2 phosphorylation, which subsequently reduces α7nAChR expression (Yi et al., 2018). Another study revealed that SIN disrupts p-ERK and p-P38, thereby regulating neutrophil migration and contributing to the alleviation of RA (Jiang et al., 2023). Additionally, it is well established that CD11b and vascular cell adhesion molecule-1 (VCAM-1) are regulators of cell migration and adhesion. A separate study indicated that Antcin K inhibits MEK1/2-ERK activity, resulting in decreased CD11b expression in monocytes and reduced VCAM-1 expression in FLS, thereby inhibiting the migration and adhesion of FLS and monocytes to the inflammatory site (David et al., 2022).

In addition to the above natural compounds, which play therapeutic roles in RA, jatrorrhizine hydrochloride, tomatidine, magnoflorine, sclareol, stauntoside B, and total flavonoids of Rhizoma Drynariae have also been reported to modulate the MAPK/ERK pathway, thereby alleviating the pathology of RA. Jatrorrhizine hydrochloride is an active ingredient extracted from Coptis chinensis, with antitumour, antibacterial and anti-inflammatory effects (Slobodníková et al., 2004; Liu et al., 2013). In MH7A cells, jatrorrhizine hydrochloride can inhibit the inflammatory response through the inhibition of ERK and p38 phosphorylation, thereby exerting a potent inflammatory modulating effect in RA (Qiu et al., 2018). Tomatidine is a natural steroidal alkaloid with immunomodulatory, antiviral and antitumour activities (Diosa-Toro et al., 2019; Mukherjee et al., 2023; Xu et al., 2024a). In CIA rat models, tomatidine has been shown to inhibit the production of IL-6, IL-1β, and TNFα by modulating the MAPK/ERK pathway (Xiaolu et al., 2021). Similarly, in CIA mouse models, both magnoflorine and sclareol have been demonstrated to reduce inflammatory mediators and mitigate the severity of RA by inhibiting p-ERK (Sen-Wei et al., 2018; Lei et al., 2023). Additionally, stauntoside B, a class of steroidal glycoside compounds, inhibits MAPK/ERK activation and therefore suppresses macrophage activation to produce inflammatory factors (Jianxin et al., 2016). Chen et al. (2022) found that total flavonoids of Rhizoma Drynariae alleviate inflammation and synovial abnormalities in CIA rats and reduce the inflammatory response of FLS cells by inhibiting the abnormal activation of the MAPK/ERK pathway (Table 2).

Table 2 Natural compounds modulating MAPK/ERK activity.

Natural Compound	In vivo/in vitro model	Mechanism	Function	Reference	
Imperatorin	FLS	Inhibit MAPK/ERK activation	Inhibiting the excessive proliferation and migration of FLS cells	Wei et al. (2022)	
Naringenin	FLS	Inactivation of MAPK/ERK pathway decreased MMPs	Promote apoptosis of FLS cells	Yirixiati et al. (2021)	
Sinomenine	FLS/ neutrophil	Inactivation of ERK1/2 pathway decreased Egr-1 and α7nAChR	Mitigate synovial inflammation, FLS proliferation and neutrophil migration	Yi et al. (2018) Jiang et al. (2023)	
Antcin K	FLS and monocyte	Inhibit MEK1/2-ERK activity	Inhibit the migration and adhesion of FLS and monocyte	David et al. (2022)	
Jatrorrhizine hydrochloride	MH7A cell	Inhibit the ERK pathway	Inhibit the inflammatory response	Qiu et al. (2018)	
Tomatidine	CIA rat	Inactivation of MAPK/ERK pathway decreased IL-1β, IL-6 and TNF-α	Anti-inflammatory	Xiaolu et al. (2021)	
Magnoflorine	CIA mice	Inhibit ERK protein phosphorylation	Anti-inflammatory	Lei et al. (2023)	
Sclareol	CIA mice	Inhibit MAPK/ERK activation	Anti-inflammatory	Sen-Wei et al. (2018)	
Stauntoside B	Macrophage	Inhibit MAPK/ERK activation	Suppress macrophage activation to produce inflammatory factors	Jianxin et al. (2016)	
Total flavonoids of Rhizoma Drynariae	CIA rat/FLS	Inhibit the abnormal activation of the MAPK/ERK pathway	Alleviate inflammation and synovial abnormalities in CIA rats and reduce the inflammatory response of FLS cells	Chen et al. (2022)	

Current studies have shown that the above compounds can inhibit specific pathological processes of RA by targeting the MAPK/ERK pathway. However, certain natural compounds, because of their complex or polar structures, have low absorption and utilization within the body and may contain toxic side effects. Therefore, questions have been raised regarding their safety and practicality. Thus, further preclinical and clinical trials are necessary to confirm the efficacy and safety of these natural substances.

Biomolecular approaches in targeting MAPK/ERK for RA treatment

Sprouty2 is a protein that negatively regulates signaling pathways and is involved in cell differentiation, growth, and migration (Zhang et al., 2021). Sprouty2 is a potent inhibitor of the Ras/ERK pathway, which directly combines with the Raf kinase to regulate the MAPK/ERK (Brady et al., 2009). It has been proposed that overexpression of Sprouty2 reduces TNFα-stimulated ERK phosphorylation in FLS by interfering with Raf activity, subsequently leading to a decrease in the synthesis of MMPs, as well as cell proliferation (Wei et al., 2015).

Dual Specificity Phosphatase 5 (DUSP5) inhibits ERK pathway by dephosphorylating ERK to inhibit its nuclear translocation, thereby regulating cell proliferation and differentiation (Ferguson et al., 2013; Buffet et al., 2017). DUSP5 was demonstrated to promote osteoblast differentiation, and its overexpression via tail vein injection effectively reversed bone loss in mice (Liu et al., 2021). In contrast, knockdown of DUSP5 enhanced activation of the ERK pathway and expression of IL-1β-induced MMPs and inducible nitric oxide synthase (iNOS), while simultaneously decreasing the expression of IL-10 and tissue inhibitor of metalloproteinases 3 (TIMP3), which contributes to anti-inflammation (Wu et al., 2020). In RA, overexpressed DUSP5 inhibited the migration, growth, and invasion of MH7A by inactivating ERK (Rongrong et al., 2022). Administration of pcDNA-DUSP5 to CIA mice maintained the balance between Treg and Th17 cells by inhibiting p-ERK activity, thereby attenuating the inflammatory response in arthritis (Moon et al., 2014). This suggests that DUSP5 may act as a potent inhibitor of MAPK/ERK, attenuating erosive damage to articular cartilage by inhibiting the activity of MAPK/ERK in RA, thereby reducing joint destruction and deformation.

Melatonin is an important neuroendocrine hormone, belonging to the class of biogenic amines. It has strong antioxidant activity and immunomodulatory effects (Reiter et al., 2016; Bagherifard et al., 2023). Melatonin is involved in regulating inflammation and cell signaling during the progression of RA. Oral administration of melatonin to CIA mice significantly improved arthritic swelling (Su et al., 2024). Furthermore, melatonin can inhibit the growth of FLS by inducing the phosphorylation of ERK to upregulate P27 and P21 (Seong-Su et al., 2009).

In addition to the above biomolecules, some peptides may also ameliorate RA by mediating MAPK/ERK signaling pathway. For instance, nesfatin-1 is a significant endogenous neuropeptide. In rats with adjuvant-induced arthritis (AIA), arthritis symptoms were alleviated following intra-articular injection of nesfatin-1. The underlying mechanism involves nesfatin-1’s ability to reduce ASIC1a protein through the MAPK/ERK, thereby mitigating acid-induced inflammatory responses and oxidative stress (Xu et al., 2022) (Table 3).

Table 3 Biomolecules regulate the MAPK/ERK signaling pathway.

Biomolecules	In vivo/in vitro model	Mechanism	Function	References	
Sprouty2	FLS	Bind to the Raf kinase to regulate the MAPK pathway, inhibit MMPs and pro-inflammatory cytokines	Anti-inflammatory and inhibit proliferation of FLS	Wei et al. (2015)	
DUSP5	MH7A/CIA mice	Inhibit p-ERK nuclear translocation and maintain the balance between Treg and Th17 cells	Inhibit the growth, migration and invasion of MH7A and reversed bone loss in mice	Wu et al. (2020) Rongrong et al. (2022) Moon et al. (2014)	
Melatonin	FLS/CIA mice	Inactivation of MAPK/ERK pathway decreased P21 and P27	Inhibit the proliferation of FLS	Seong-Su et al. (2009)	
Nesfatin-1	AIA rat/chondrocyte	Inactivation of MAPK/ERK pathway decreased ASIC1a	Anti-inflammatory	Xu et al. (2022)	

Although some studies have suggested that biomolecules inhibiting the MAPK/ERK may play a role in RA, their clinical safety and efficacy have yet to be further validated. Given that RA is a complex autoimmune disease, single-targeted biomolecule therapy may not be able to achieve the desired clinical effects and may also involve certain risks. Therefore, it is essential to conduct more extensive animal experiments and clinical trials to comprehensively assess the potential and safety of these biomolecules in the treatment of RA. Additionally, exploring multi-targeted and multi-pathway integrated treatment strategies is necessary to provide RA patients with more effective and safer therapeutic options. The abbreviations are shown in Table 4.

Table 4 The following abbreviations are used in this manuscript.

RA	Rheumatoid arthritis	
MAPK/ERK	Mitogen-activated protein kinases/extracellular regulated protein kinases	
PI3K/AKT	Phosphoinositide 3-kinase/protein Kinase B	
JAK/STAT	Janus kinase/signal transducer and activator of transcription	
NF-κB	Nuclear transcription factor-κB	
SHH	Sonic hedgehog	
MEK	MAPK/ERK kinase	
FLS	Fibroblast-like synoviocytes	
ASIC1a	Acid-sensing ion channel 1a	
HIF-1α	Hypoxia-inducible factor 1α	
CIA	Collagen-induced arthritis	
Dyrk1A	Dual-specificity tyrosine-regulated kinase 1A	
GZMK	Granzyme K	
miR	MicroRNA	
LPA	Lysophosphatidic acid	
MMPs	Matrix metalloproteinases	
ICAM-1	Intercellular adhesion molecule-1	
PCSK6	Proprotein convertase subtilisin/kexin type 6	
VEGF	Vascular endothelial growth factor	
EPC	Endothelial progenitor cell	
HUVEC	Human umbilical vein endothelial cells	
Fkn	Fractalkine	
PGE2	Prostaglandin E2	
SMO	Smoothened	
AT	Atorvastatin	
SIN	Sinomenine	
Egr-1	Early growth response gene-1	
VCAM-1	Vascular cell adhesion molecule-1	
DUSP5	Dual Specificity Phosphatase 5	
iNOS	Inducible nitric oxide synthase	
TIMP3	Tissue inhibitor of metalloproteinases 3	
AIA	Adjuvant-induced arthritis	

Conclusions

Despite existing research highlighting the role of the MAPK/ERK pathway in the pathology of RA, numerous unresolved issues persist regarding the relationship between RA and this pathway. First, most studies on MAPK/ERK pathway-targeted therapies for RA remain at the cellular or small-animal research stage, lacking large-scale clinical trials. Second, there is significant variability in patients’ responses to these therapies, complicating the development of standardized treatment protocols. Furthermore, although several drugs aimed at inhibiting the MAPK/ERK pathway have progressed to the preclinical research stage and demonstrate potential for treating RA, these drugs frequently fail to achieve precise inhibition of this signaling pathway. Moreover, inhibiting the MAPK/ERK pathway may lead to unforeseen side effects. The efficacy and safety of MAPK/ERK pathway-targeted therapies in clinical applications require further validation. Although some targeted drugs have demonstrated effects in initial studies, subsequent investigations often failed to replicate these results. Currently, there is a lack of sufficient long-term clinical data regarding the effects of prolonged inhibition of the MAPK/ERK pathway on other physiological aspects and the immune system, particularly concerning immune function. To address these challenges, future research should prioritize a more in-depth exploration of underlying mechanisms, the use of alternative animal models (pigs, monkeys, or dogs), the optimization of targeted therapy strategies, and the investigation of personalized treatment approaches. Through multi-center, long-term studies, we may gain a clearer understanding of the role of the MAPK/ERK pathway in RA and refine treatment regimens. In addition, the combination of MAPK/ERK-targeted drugs with other treatment modalities, or the application of gene editing technologies to target critical genes within the MAPK/ERK pathway, may emerge as a pivotal strategy in future clinical research, significantly enhancing patient clinical outcomes.

In summary, the activation of the MAPK/ERK pathway increases the levels of various chemokines and inflammatory genes, influencing pathological processes such as inflammation, cell growth, cell migration, and angiogenesis. Regulating the MAPK/ERK pathway can effectively inhibit these pathological responses, thereby improving the condition of RA patients. A more comprehensive understanding of its complex regulatory mechanisms will facilitate the development of more precise and effective treatment strategies.

We appreciate Ziyu Zhou, Yunjie Hu for their valuable suggestions on this manuscript.

Additional Information and Declarations

Competing Interests

The authors declare that they have no competing interests.

Author Contributions

Jun Xie conceived and designed the experiments, performed the experiments, analyzed the data, prepared figures and/or tables, authored or reviewed drafts of the article, and approved the final draft.

Sijuan Sun conceived and designed the experiments, performed the experiments, prepared figures and/or tables, authored or reviewed drafts of the article, and approved the final draft.

Qingzhou Li conceived and designed the experiments, performed the experiments, prepared figures and/or tables, authored or reviewed drafts of the article, and approved the final draft.

Yuhui Chen performed the experiments, analyzed the data, authored or reviewed drafts of the article, and approved the final draft.

Lijun Huang analyzed the data, prepared figures and/or tables, authored or reviewed drafts of the article, and approved the final draft.

Dong Wang conceived and designed the experiments, authored or reviewed drafts of the article, and approved the final draft.

Yumei Wang conceived and designed the experiments, performed the experiments, authored or reviewed drafts of the article, and approved the final draft.

Data Availability

The following information was supplied regarding data availability:

This is a literature review.

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
