# Peer review of "MAPK/ERK signaling pathway in rheumatoid arthritis: mechanisms and therapeutic potential"

_PeerJ, doi:10.7717/peerj.19708_

## Round 0.1 · original submission · Minor Revisions

Please address comments of both reviewers and submit your revised manuscript along with point-wise responces.

·

Basic reporting

This review provides a comprehensive analysis of the MAPK/ERK signaling pathway in the pathophysiology of rheumatoid arthritis (RA) and its therapeutic potential. The topic is clinically relevant and timely, given the unmet need for novel RA therapies. The authors highlight the under-reviewed role of MAPK/ERK in RA compared to other MAPK pathways (e.g., p38 and JNK) and emphasize its synergistic interactions with other key pathways (e.g., NF-κB, JAK/STAT, SHH, PI3K/AKT). The focus on molecular mechanisms, cross-pathway crosstalk, and therapeutic strategies offers valuable insights for both basic and translational research. The manuscript is well-structured, concise, and clearly outlines the review’s objectives.

Experimental design

no comment

Validity of the findings

no comment

Additional comments

However, this manuscript should undergo some minor revisions before publication. The manuscript should briefly specify how MAPK/ERK drives RA pathology (e.g., synovial fibroblast activation, cytokine production, osteoclastogenesis) and highlight key molecular players (e.g., Ras, Raf, MEK, ERK kinases). A schematic diagram in the full manuscript illustrating MAPK/ERK crosstalk with other pathways (e.g., NF-κB) would strengthen mechanistic clarity. The author still overlooked some literature in this field, such as in the section on "Natural compounds modulating MAPK/ERK activity". “Network Pharmacology Analysis and Experimental Validation to Investigate the Mechanism of Total Flavonoids of Rhizoma Drynariae in Treating Rheumatoid Arthritis.” The manuscript introduces that Total Flavonoids of Rhizoma Drynariae alleviate RA through the MAPK signaling pathway, but it does not appear in this chapter.

Reviewer 2 ·

Basic reporting

The review is about the MAPK/ERK signaling pathway in rheumatoid arthritis, focusing on the pathway’s function and mechanism of action, correlation with other pathways, and potential therapeutic applications of targeting this pathway. The author did a good job explaining the implications and importance of targeting this pathway to counter RA.
The article is written in professional English and is easy to follow and comprehend.
The introduction properly addressed the reason for writing this review with adequate background information. The review is timely and written to address the gaps, and it will provide the audience enough understanding to grasp a complete view of the topic.

Experimental design

The review is structurally well formulated, and the writing flow is easy to follow.

Validity of the findings

The reported findings of this review will benefit the field’s advancement. The conclusion identified the unresolved questions with necessary future directions.

Additional comments

The article can be accepted with the following corrections-
General Comments
In all figures and Table 4, capitalize single words (i.e., Nucleus) and the first word of Joint words (i.e., Transcription factor) and so on. In some cases, it was capitalized, and in most cases, the first word is written in lowercase.
Figure 1- please add the causal agents for stimulations instead of just writing stimulation.
Line 43, rewrite the sentence, starting with the clinical manifestations.
Line 65-66, the indicated reference doesn’t mention anything about RA or MAPK/ERK pathway’s relevance to RA. Please check the reference or add the one that addresses the claim as the most classical and central to the disease.
Line 120, ERK1 and ERK2, ERK2 is made up of 360AA. Please indicate that or add the AA number for both.
Line 194, add FLS full form where it is first introduced.
Line 229-231, please rewrite the sentence as it seems confusing, and if possible, indicate which types of cells.
Line 374, the word optimisation should use Z, as in other cases, written as optimization.
Physiological functions of the MAPK/ERK pathway
Please expand this point, as in multiple cases, especially for drugs, it was written can affect the normal physiology; a more detailed function of this pathway will give readers an understanding of why targeting this pathway is risky.

---

## Round 0.2 · accepted · Accept

The authors have addressed all of the reviewers' comments and this manuscript is now ready for publication.

·

Basic reporting

no comment

Experimental design

The design of the article is very reasonable.

Validity of the findings

No concerns regarding the validity of the findings

Additional comments

The revised manuscript is now suitable for publication.

Reviewer 2 ·

Basic reporting

The authors addressed all the questions and made necessary changes where needed.

Experimental design

N/A

Validity of the findings

N/A

Additional comments

N/A